# Interaction between Risk Single-Nucleotide Polymorphisms of Developmental Dyslexia and Parental Education on Reading Ability: Evidence for Differential Susceptibility Theory

**DOI:** 10.3390/bs14060507

**Published:** 2024-06-19

**Authors:** Qing Yang, Chen Cheng, Zhengjun Wang, Ximiao Zhang, Jingjing Zhao

**Affiliations:** School of Psychology, Shaanxi Normal University, 199 South Chang’an Road, Xi’an 710062, China; yangq1110@163.com (Q.Y.); 23121110609@stu.xidian.edu.cn (C.C.); zhling307@126.com (Z.W.); zhangxm@snnu.edu.cn (X.Z.)

**Keywords:** gene–environment interaction, reading ability, differential susceptibility, re-parameterized regression

## Abstract

While genetic and environmental factors have been shown as predictors of children’s reading ability, the interaction effects of identified genetic risk susceptibility and the specified environment for reading ability have rarely been investigated. The current study assessed potential gene–environment (G×E) interactions on reading ability in 1477 school-aged children. The gene–environment interactions on character recognition were investigated by an exploratory analysis between the risk single-nucleotide polymorphisms (SNPs), which were discovered by previous genome-wide association studies of developmental dyslexia (DD), and parental education (PE). The re-parameterized regression analysis suggested that this G×E interaction conformed to the strong differential susceptibility model. The results showed that rs281238 exhibits a significant interaction with PE on character recognition. Children with the “T” genotype profited from high PE, whereas they performed worse in low PE environments, but “CC” genotype children were not malleable in different PE environments. This study provided initial evidence for how the significant SNPs in developmental dyslexia GWA studies affect children’s reading performance by interacting with the environmental factor of parental education.

## 1. Introduction

Developmental dyslexia (DD) is known to be a hereditary neurological disorder with an unbalanced development between reading ability and age, which cannot be accounted for by low intelligence, inadequate education, visual or auditory acuity deficits, or other neurodevelopmental disorders [1]. The prevalence of DD is reported to be around 1.3–17.2% among school-age students varying in different orthographies and depending on the criteria used for diagnosis [2], and its etiology is influenced by both genetic and environmental factors [3].

Numerous studies have also suggested that environmental factors, such as birth weight, born preterm, home literacy environment (HLE), socioeconomic status (SES), and parental education (PE), have important influences on the development of children’s reading ability [4,5,6,7,8]. The studies have highlighted the importance of family socioeconomic status in early childhood literacy development, and these findings consistently show that children with a high socioeconomic status have higher reading and language skills before and after formal education than those with a low socioeconomic status [9,10,11].

G×E on twin studies have been well-studied in recent years. For example, behavioral genetic studies with identical and fraternal twins have suggested that the degree of genetic influence on, or heritability of, individual differences in cognitive and reading abilities varies with SES [12] or PE [7]. The bioecological model is usually applied in twin studies to explain how the heritability of certain phenotypes varies with the environment. Kremen et al. (2005) first investigated this theory in middle-aged twins, and found that the heritability of reading recognition ability during midlife was higher in individuals due to the decreasing variance in common environmental influences with increasing PE [13]. Further evidence supported that genetic influence was higher among children whose parents had a high level of education than those children whose parents had a lower level of education [7]. G×E effects are explained not only by the bioecological model among twins but also by the diathesis–stress model among unrelated individuals (illustrated later in the Introduction).

Molecular genetic studies further reported that the influence of dyslexic candidate genes on reading disability may depend on the specific environmental influence [14]. The diathesis–stress model suggested that the effect of the risk allele for a particular behavior would be worse in poor environments and the risk allele would not benefit from more supportive environments. The model has been proposed to explain why certain behavioral disorders had a greater association with risk genes in environments [15,16]. Mascheretti et al. (2013) explored the interactions of five candidate genes (DYX1C1, DCDC2, ROBO1, KIAA0319, and TTRAP) with a series of environmental factors through unrelated probands and siblings [17]. The results showed that children with DYX1C1 risk alleles would perform worse in adverse environments, such as low birth weight, low SES, and history of smoking during pregnancy, but would not be affected in positive environments [17]. This model equally explained G×E on orthography processing in Chinese children, i.e., children carrying the minor allele of rs1091047 exhibited a smaller N170 effect, which is activated in visual words and thus influences reading abilities [18], from low-home-literacy environments than those from high-home-literacy environments [19].

Recently, however, a theoretical alternative to the diathesis–stress model has been proposed (i.e., the differential susceptibility model) and applied to the study of gene–environment interactions [20]. Consistent with the perspective of diathesis–stress, in poor environments, individuals with a putative high-risk allele will exhibit poorer outcomes, and in positive environments, individuals with a putative high-risk allele will show better outcomes compared with those with the low-risk allele [21]. In the area of early literacy instruction, Kegel et al. (2011) provided evidence that children had differential susceptibility in an environment. Individuals with the 7-repeat allele of the dopamine receptor D4 (DRD4-7R) profited most from the positive feedback of computer programs, whereas they performed worse in early literacy skills in the absence of such feedback [22]. And more recently, many other molecular genetic studies of gene–environment interactions have confirmed this method [23,24].

In recent years, genome-wide association analysis studies (GWASs) have been used to identify single-nucleotide polymorphisms (SNPs) associated with reading or dyslexia [25,26,27,28,29,30,31,32,33,34] and four genome-wide significant variants were found among GWASs on reading abilities [28]. However, genome-wide association analysis ignored that gene expressions can be affected by environmental factors, thus genome-wide by environment interaction studies (GWEISs), a novel method, are used to discover the SNPs under environmental effects. GWEISs have not been studied on reading abilities because the sample size is usually four times larger than the sample size needed to discover the main effect of genes [35]. Distinguishing from G×E studies on SNPs from candidate genes, our aim is to tentatively investigate the feasibility of the interaction between SNPs from GWASs and the environment to provide evidence for GWEISs on reading abilities.

The nine SNPs nominally associated with reading traits were selected from previous GWASs in different cohorts in developmental dyslexia and reading ability as genetic factors in the reading ability of Chinese children. The cumulative genetic effect of these nine SNPs was also explored for increasing the genetic effect due to the small effect of one SNP. Therefore, we examined the interactions between the cumulative genetic effect and single SNPs and environmental factors. We selected parental education (PE), which has been shown to be a strong predictor for a variety of cognitive outcomes in children [36,37], as a proxy of SES. To further consider PE as an environmental variable, a correlation analysis was conducted. We predicted that if genetic and environmental influence were interdependent in reading ability, the effects of the genetic signal would vary along the PE distribution. In confirmation analysis, two gene–environment interaction models (the diathesis–stress model vs. the differential susceptibility model) were compared in a re-parameterized analysis to identify which model was a better fit for our data.

## 2. Methods

### 2.1. Participants

A total of 3217 primary school students from grade 3 to grade 6 were recruited from three provinces in the northwestern part of China (Shaanxi Province, Gansu Province, and Inner Mongolia Province). All these participants were school-aged children without any neurological disorder or history of taking neurotropic drugs. In total, 2476 saliva samples of these participants were eligible for subsequent gene assaying and 2415 students’ genetic data were available at last. Finally, there were 1477 students who provided all the phenotype, genotype, and environmental data (age = 116.34 ± 12.14 months, male/female = 737/740). Ethical approval was obtained from Shaanxi Normal University and written informed consent was obtained from all the participants’ parents.

### 2.2. Measures

*Genetic analyses.* DNA was obtained from oral epithelial cells in students’ saliva samples and 2476 samples were genotyped using the Illumina Asian Screening Array (650 K) by Beijing Compass Biotechnology (Beijing, China). Quality control was performed for standard quality control metrics [38,39] using PLINK v1.9 (http://pngu.mgh.harvard.edu/purcell/plink/ (accessed on 1 June 2020)). Eight samples were excluded as they had sex discrepancies between the records and the genetically inferred data. Fifty-three samples were removed because they had unexpected duplicates or probable relatives (PI-HAT > 0.20). Next, SNPs were eliminated if they showed a variant call rate < 0.95, a minor allele frequency (MAF) < 0.02, a missing genotype data (mind) < 0.90, or a Hardy–Weinberg equilibrium (HWE) < 10^−5^ with each dataset. Then, autosomal variants were aligned to the 1000 G genomes phase 1v3 reference panel for imputation, which followed the standard procedure consistent with previous GWASs (see [27]). Finally, significant reading-related or dyslexia-related SNPs were extracted from our data.

We collected all the SNPs that were associated with reading and recorded by the GWAS catalog (https://www.ebi.ac.uk/gwas/ (accessed on 1 June 2020.)). There were 4 SNPs (rs1555839, rs17663182, rs349045, and rs16928927) that showed a strict significant association with reading on a genome-wide level (*p* ≤ 5 × 10^−8^); however, none of them existed in the current study sample. Thus, we loosened the criteria from *p* ≤ 5 × 10^−8^ to *p* ≤ 5 × 10^−7^. Then, there were 22 SNPs recorded by the GWAS catalog (for detailed information see Appendix A) and 9 of them existed in our samples (see Table 1).

*Reading ability*. Each child’s reading ability was tested using the *Chinese character recognition test*, a reading test used in Mainland China [40]. The test consisted of 150 single Chinese characters selected from China’s *Elementary School Textbooks* (1996). The average frequency of the characters was 182 per million (ranging from 0 to 2282), and the reliability of this test was 0.95 [40]. Each child was individually tested and required to read aloud each character at a time. Each child’s reading score, namely the number of characters read correctly, was recorded. Finally, 2269 children completed this test.

*Parental education (PE).* Students’ PE (parental education) was gathered through questionnaires. The scores ranged from 1 to 8, representing the highest educational qualifications of their parents, from primary school, junior high school, senior high school, or junior college to an undergraduate, master’s, doctoral, or post-doctoral degree. Both mother’s and father’s education levels were taken into account, with the PE being the average of the two. In total, data on PE of 1507 students were collected.

### 2.3. Data Analysis

Both exploratory and confirmatory analytic approaches consistent with Widaman et al. (2012) were used for 9 SNPs [21]. Standard exploratory analysis was aimed at testing whether there were G×E effects on character recognition. A confirmatory re-parameterized approach was employed to contrast the different hypotheses of G×E, i.e., strong and weak forms of the differential susceptibility and diathesis–stress models to determine which provided the best, most parsimonious fit to the data.

*Exploratory single SNP.* We tested the gene–environment interactions for each of the nine SNPs with the environment. The standard multiple regression model is similar to Equation (1):*Y* = *A*_0_ + *A*_1_*X*_1_ + *A*_2_*T* + *A*_3_(*X*_1_ × *T*) + *A*_4_·*Age* + *A*_5_·*Sex* + *E*(1)
where *Y* is the dependent variable (reading ability, i.e., characters recognition scores); *X*_1_ is the environment variable (PE); *T* is trivariate (0 represented none-risk alleles, 1 represented 1 risk allele and 2 represented 2 risk alleles); *X*_1_ × *T* is the product variable of the gene–environment interaction; *A*_1_ and *A*_2_ are regression slopes for main effects of the environment (*X*_1_) and SNP (*T*), respectively; *A*_3_ is the regression coefficient for the product variable (*X*_1_ × *T*) and represents the difference in slope on *X*_1_ among the “zero risk allele” group, “one risk allele group”, and “double risk alleles group”; *A*_0_ is the intercept; *A*_4_ and *A*_5_ are the regression slopes for covariant age and sex, respectively; and E is a stochastic error term.

Equation (1) is fit once excluding the product term (*X*_1_ × *T*), testing the main effects of SNPs and PE (Model 1). The significant increase in the squared multiple correlation, *R*^2^, after adding the product term, provides evidence for G×E interactions (Model 2).

*Exploratory CGS.* We further computed the cumulative genetic score (CGS) by adapting from known studies to create risk scores (see Table 1 Beta), with the risk allele coded as 1 and the normal allele coded as 0 (e.g., if the two different alleles of rs281238 were C and T, and the risk allele was T, then the genotypes of CC, CT, and TT will be coded as 0, 1, and 2, respectively). Parental education (PE) was the environmental variable *X*_1_, and a continuous variable CGS was the genetic variable *X*_2_. Age and sex were covariates. The standard multiple regression model can be written as:*Y* = *A*_0_ + *A*_1_*X*_1_ + *A*_2_*X*_2_ + *A*_3_(*X*_1_ × *X*_2_) + *A*_4_·*Age* + *A*_5_·*Sex* + *E*(2)
where *Y* is the dependent variable (reading ability); *A*_0_ is the intercept; *A*_1_ and *A*_2_ are regression slopes for main effects of the environment (*X*_1_) and CGS (*X*_2_), respectively; *A*_3_ is the regression coefficient for the product variable (*X*_1_
*× X*_2_) and represents the difference in slope on *X*_1_ for the CGS; *A*_4_ and *A*_5_ are the regression slopes for covariant age and sex; and *E* is a stochastic error term.

*Confirmatory SNPs.* Following Widaman et al. (2012), we re-parameterized the regression model, allowing for the testing of alternative forms of the G×E interaction [21], as:(3)$$Y=\left\{\begin{array}{c} T=0\;\;Y=A_{0}+A_{1}(X_{1}-C)+A_{4}\cdot Age+A_{5}\cdot Sex+E\\ T=1\;\;Y=A_{0}+A_{2}(X_{1}-C)+A_{4}\cdot Age+A_{5}\cdot Sex+E\\ T=2\;\;Y=A_{0}+A_{3}(X_{1}-C)+A_{4}\cdot Age+A_{5}\cdot Sex+E \end{array}\right.$$

Here, *C* is the point on *X*_1_. If the cross point of *C* and its confidence interval (CI) is within the range of value on *X*_1_ observed in this study, the interaction tested is disordinal, reflecting the differential susceptibility model. On the contrary, if the cross point of *C* or its confidence interval (CI) is greater than or equal to the most positive point on *X*_1_ in this study, the interaction is ordinal, consistent with the diathesis–stress model.

Equation (3) is the re-parameterized regression model for Equation (1); *C* is the point on *X* at which the slopes for the different genotype groups cross. If the cross point of *C* and its confidence interval (CI) are within the range of values on the environment, the interaction tested is disordinal, reflecting the differential susceptibility model. Conversely, if the cross point of *C* or its confidence interval (CI) is greater than or equal to the most positive point on the environment in this study, the interaction is ordinal, consistent with the diathesis–stress model.

Next, to compare the efficiency of the strong or weak differential susceptibility model and diathesis–stress model, we construct Model 3a, Model 3b, Model 3c, and Model 3d. In Model 3a and Model 3b, we assume that the cross-over point C fell at the range of *X* observed values and the G×E is disordinal. If the slope of the none-risk allele genotype group (*T* = 0) is fixed at zero (i.e., *A*_1_ = 0), the model (Model 3a) in Equation (3) is consistent with the strong differential susceptibility model, which means that the none-risk allele group is unaffected by the environment. Relaxing the constraint that *A*_1_ = 0 leads to Model 3b, consistent with the weak differential susceptibility model, means the slope for the none-risk allele genotype group differs significantly from zero. If we could add a free parameter in Equation (3) that can explain significantly more variance than Model 3a, we would accept Model 3b; otherwise, we would accept Model 3a.

For Model 3c and Model 3d, we assume that *C* is the max(*X*), namely Mean(*X*) + 3 SD. If A1 is fixed and the environment has no effect on the none-risk allele group, the model would conform to a strong diathesis–stress model, Model 3c. If the none-risk allele genotype group is affected by the environment (i.e., *A*_1_ ≠ 0), the model would reflect the weak diathesis–stress model, the Model 3d. We would reject Model 3d if there is no significant increase in *R*^2^ compared with Model 3c. Then, an ANOVA (Analysis of Variance), the Akaike information criterion (AIC), and the Bayesian information criterion (BIC) are employed to evaluate the applicability between these four models. For both the AIC and BIC, the lower the value, the more efficient the model.

*Confirmatory CGS.* We constructed the re-parameterized equation of the linear × linear interaction [21] as:*Y* = *A*_0_ + *A*_1_(*X*_1_ − *C*) + *A*_2_((*X*_1_ − *C*) *× X*_2_) + *A*_3_·*Age* + *A*_4_·*Sex* + *E*(4)

To compare the efficiency of the differential susceptibility model and diathesis–stress model on CGS, we construct Model 3e and Model 3f. In the model 3e, *C* is the unfixed cross-over point of CGS on *X*_1_, while in Model 3f, *C* is fixed at the maximum (*X*_1_), meaning that the cross-over point fell at the highest value for the environment observed in the sample. Through ANOVA analysis between the two models, Model 3e would be accepted if there is a significant increase in *R*^2^ due to the increased parameter *C*; conversely, we would reject Model 3e. The Akaike information criterion (AIC) and Bayesian information criterion (BIC) are employed to evaluate the applicability of these two models. For both the AIC and BIC, the lower the value, the more efficient the model.

## 3. Results

All the nine SNPs that existed in our sample conformed to Hardy–Weinberg equilibrium (*p* > 0.05, see Table 1).

### 3.1. Correlation Analysis

To exclude the gene–environment correlations, a correlation analysis was conducted (see Appendix A). The relation between CGS and PE was not significant (r_p_ = 0.001), and thus PE can be used as an environmental variable in our study. However, rs1541518 was marginally significantly correlated with PE (r_p_ = 0.005, *p* = 0.04), and we excluded this SNP in later analysis.

### 3.2. Standard Exploratory Analysis

*Single SNP.* After testing the G×E of CGS, we verified the interactions of each single SNP (Appendix A). Only the SNP of rs281238 was significant after Bonferroni correction as shown in Table 2. Results of the other SNPs are shown in Appendix A. Model 1 in Table 2 has an *R*^2^ = 0.2531 (*p* < 0.001), with a significant main effect of PE, $\hat{A}$_1_
*=* 1.88 (*SE* = 0.43), *p* < 0.001, but not the SNP main effect, $\hat{A}$_2_ = 0.01 (*SE* = 0.67), *p* = 0.98. The G×E interaction produced a significant Δ*R*^2^ of 0.0035, *p* = 0.005 (see Model 2). The coefficients of rs281238, $\hat{A}$_2_
*=* −5.08 (*SE* = 1.94), *p* = 0.009, the G×E product term, $\hat{A}$_3_
*=* −1.56 (*SE* = 0.56), *p* = 0.005, and $F\frac{Model\;1}{Model\;2}$ = 7.81, by ANOVA analysis, allowed for confirmatory re-parameterized analysis to be estimated. Figure 1A shows the difference in the impact of environment on character recognition of different genotypes.

*CGS.* The standard multiple regression of CGS and PE with G×E was fit to data on character recognition. It had an *R*^2^ = 0.2552 (*p* < 0.001), with a significant environment main effect, $\hat{A}$_1_ = 1.87 (*SE* = 0.43), *p* < 0.001, but no significant gene main effect, $\hat{A}$_2_ = −0.80 (*SE* = 0.29), *p* = 0.78. The main effects of covariate age and sex were also significant. Adding the G×E product term to the equation results in an increase in Δ*R*^2^ of 0.0023, as was the coefficient for the G×E product term itself, $\hat{A}$_3_ = 0.52 (*SE* = 0.24), *p* = 0.0307 (see Table 3 and Figure 2). After ANOVA analysis, $F\frac{Model\;1}{Model\;2}$ = 4.68, the *F* ratio > 1.0 suggested that we can further evaluate competing theoretical models [41].

### 3.3. Confirmatory Re-Parameterized Analysis

Different versions of Equation (2) reflect strong and weak forms of differential susceptibility and diathesis–stress models for G×E interaction on character recognition.

#### 3.3.1. Differential Susceptibility vs. Diathesis–Stress Model for rs281238

The estimated cross-over point *C* fell close to the sample mean on PE and the 95% CI of *C* fell within the range of PE. In Model 3c, $\hat{C}$
*=* 3.21 (*SE =* 0.36), the 95% CI of $\hat{C}$ [2.50, 3.91], and in Model 3d, $\hat{C}$
*=* 3.21 (*SE =* 0.42), the 95% CI of $\hat{C}$ [2.39, 4.03]. Model 3c represented the strong differential susceptibility model, suggesting that children without the risk allele T would be unaffected by PE, but children with different numbers of the risk allele T would be positively influenced by PE at different degrees. Model 3c explained a significant amount of variance in character recognition, *R*^2^ = 0.2567, *p* < 0.0001 (See Section 2 and Appendix A). The nested model of Model 3c, which relaxes the constraint that *A*_1_ = 0, leads to Model 3d, the weak differential susceptibility model. Model 3d had a modest increase in explaining character recognition variance over Model 3c, but did not reach the significant level, Δ*R*^2^ = 0.0002, *p* = 0.52. This suggested that adding a parameter *A*_1_ to the model cannot significantly improve the fitness of the model. Thus, we accept the strong differential susceptibility model and reject the weak differential susceptibility model.

In Model 3e, the strong diathesis–stress model, children without risk allele T would not be affected by PE; however, the reading ability of children with risk allele T would increase as PE improved, but would never catch up with the children without the allele T. Model 3e explained a significant amount of variance in character recognition, *R*^2^ = 0.2452, *p* < 0.0001. The nested model of Model 3e, which relaxes the constraint that *A*_1_ = 0, leads to Model 3f, the weak diathesis–stress model, which explained a large amount of additional variance over that explained by Model 3e, Δ*R*^2^ = 0.0078, *p* = 0.0001. Therefore, we found no significant basis for accepting the parsimonious Model 3c, supporting the weak diathesis–stress model as a more optimal representation of the data.

There are no statistical tests that can evaluate the efficiency of Model 3c and Model 3f because of the same degree of freedom; therefore, AIC and BIC values were employed to compare these two models. As Model 3c has lower AIC and BIC values than Model 3f, we accepted Model 3c, the strong differential susceptibility model, as the best model for the current data. In order to provide an effective way of presenting the interactions in data, we plotted predicted values under the strong differential susceptibility model. The plot of predicted character recognition score is shown in Figure 3, where the number of T alleles determines the malleability of children. Below the cross-over point (low level of PE), children’s reading ability increases from low to high in the TT, TC, and CC groups as a function of PE. However, above the cross-over point (high level of PE), the TT and TC group shows consistently higher performance than the CC group as a function of PE.

#### 3.3.2. Differential Susceptibility vs. Diathesis–Stress Model for CGS

The estimated cross-over point *C* fell close to the sample mean on PE and the 95% CI of *C* fell within the range of PE. In Model 3e, $\hat{C}$
*=* 3.34 (*SE =* 0.56), the 95% CI of $\hat{C}$ [2.24, 4.43], explained a significant amount of variance in character recognition, *R*^2^ = 0.2555, *p* < 0.0001 (see Table 3). Choosing the highest PE value as *C* leads to Model 3f, the diathesis–stress model. Model 3f, the diathesis–stress model, explained a significant amount of variance in character recognition, *R*^2^ = 0.2536, *p* < 0.0001. Compared to Model 3f, Model 3e had an additional parameter, $\hat{C}$, which significantly increased the *R*^2^, Δ*R*^2^ = 0.0019, *p =* 0.049, so Model 3e was accepted. In addition, we calculated the CGS of eight SNPs (excluding rs1541518) and found that the interaction between the CGS of eight SNPs and PE was significant and consistent with the differential susceptibility model (see Appendix A).

## 4. Discussion

Within the framework of G×E studies, we mainly focus on the interaction model. The interaction effects were consistent with the differential susceptibility model: both CGS and rs281238 interact with PE; individuals with susceptibility benefit from a high parental education level, whereas they are easily impaired by a low parental education level. To our knowledge, this is the first study to investigate the interaction between environmental factors and SNPs associated with reading-related traits from GWASs in reading ability using multiple regression and re-parameterized analysis. There was no evidence to support the diathesis–stress model in the current study as the model did not explain more variance.

More importantly, we observed that the gene–environment interactions between rs281238 and parental education on reading ability were consistent with the differential susceptibility model. Children with “TT” are more developmentally plastic or malleable [42] and they are better in high PE and worse in low PE, while children with “CC” genotypes were not susceptible to different PE. The SNP, rs281238, located in SEMA6D, is a member of the semaphorin gene family, which encodes a transmembrane protein that may play a navigational role in the signaling process of the nervous system [43]. Other SNPs in SEMA6D, such as rs1378214, rs281320, and rs281323, have been reported to be significantly associated (*p* < 5 × 10^−8^) with cognitive ability, attention deficit hyperactivity disorder (ADHD), and education attainment and have been replicated in independent studies [44,45,46,47]. SNPs in SEMA6D, such as rs12910916, rs2573570, and rs8039398, have also been reported to be associated with reading, mathematics, cognition, education achievement, and ADHD [28,46,47,48]. All these results suggest the impact of SEMA6D on cognitive and reading-related traits. Taken together, our results provide new evidence for the importance of SEMA6D in the complex area of cognition–reading ability.

We calculated the cumulative genetic scores for nine SNPs and found an interaction between CGS and parental education on reading ability, suggesting the effect of multiple genes on complex traits [49]. The study also demonstrated cumulative effects on genes [50,51], thus, those children carrying more plasticity alleles were more affected in reading than those carrying fewer, providing new evidence for the differential susceptibility model for G×E effect on reading ability. This finding also added to the evidence supporting the differential susceptibility model for the G×E effect on behavioral traits [24,52] which offers an alternative, evolutionary-inspired view: in order to enhance adaptation to different environments, some individuals are simultaneously sensitive to both negative and positive experiences [53]. Although GWAS found significant loci, their explanatory power for reading was very low. This finding supports the idea that G×E interaction is a possible aid in gene discovery [35] and the consideration of the environment in GWASs could increase the efficiency of gene discovery. The results provide supportive evidence for the polygenic approach to address the complexities of genetics, behavioral phenotypes, and environmental influences. Because children with high CGS can be affected most by PE, we proved a replicable finding that individuals with high polygenic susceptibility scores (PSSs) might be intervened successfully [42,54].

Notably, the current study has several limitations. First, genome-wide significant SNPs reported in the previous literature on reading and dyslexia [28] were not genotyped in our data and therefore were not tested for a G×E effect on reading ability in this study. Future studies with genome sequencing of this Chinese cohort may be desirable to further test the G×E effect on reading ability with genome-wide significant SNPs. Second, we only focused on parental education as an environmental factor rather than correlations between gene and environment (rGE), which means that an individual’s genotype influences, or is associated with, their exposure to the environment [55]. Randomized control trials can be used to explore G×E studies to eliminate rGE and other accurate environmental factors might be valuable to be used to test the G×E effect on reading ability in future studies [56]. Finally, to further verify the external validity of this study, it is essential to replicate these findings in independent cohorts in the future.

We must place greater emphasis on early genetic screening and, at the same time, on guiding parents to stimulate children’s reading potential. Parental education often correlates with rich cognitive resources and a rich home reading environment. Due to the existence of gene–environment interactions, educators, especially schools and teachers, can provide an environment conducive to reading development for children at risk of dyslexia, thereby reducing the incidence of dyslexia. Individuals at risk for dyslexia, whose lives may be adversely affected, can enhance their character recognition skills and, consequently, their text comprehension abilities by improving access to scarce reading resources in their environment.

In conclusion, we have provided initial evidence for how the significant SNPs in the GWA studies of developmental dyslexia affect children’s reading performance by interacting with the environmental factor of parental education. Our results indicated the validity of the differential susceptibility model in reading ability. Moreover, the current study was performed in normal school-aged students and the reading ability is normally distributed; thus, these results may provide an explanation for both children with high and low reading ability. Future gene–environment research should consider both positive and adverse environmental factors.

## Figures and Tables

**Figure 1 behavsci-14-00507-f001:**
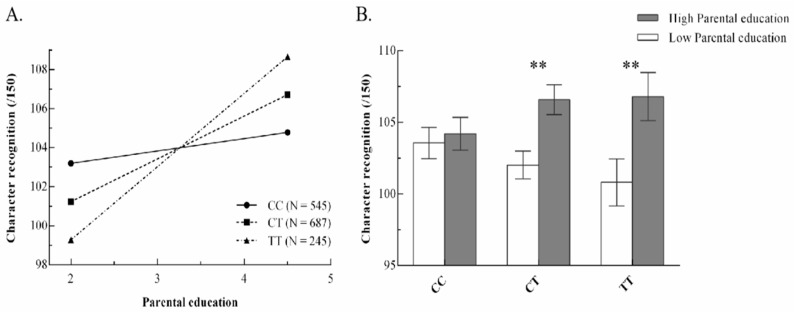
(**A**) Simple slope analysis for character recognition in different genotype groups, and (**B**) character recognition of different children’s genotypes in different environments by two-way anova (CC = 279/266, CT = 373/314, TT = 128/117), ** *p* < 0.01.

**Figure 2 behavsci-14-00507-f002:**
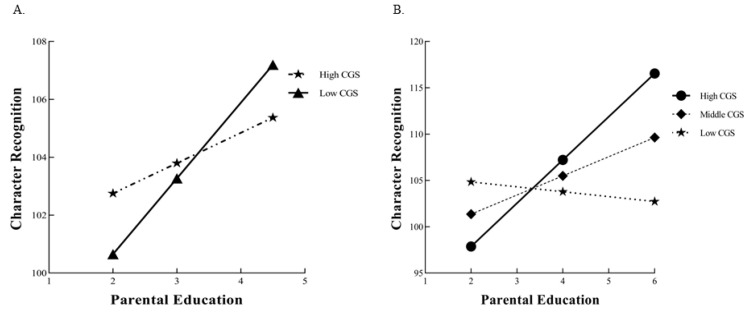
(**A**) A simple slope analysis of character recognition in low and high CGS subgroups; (**B**) the plots for the results of the interaction between CGS and PE to predict character recognition in the differential susceptibility model.

**Figure 3 behavsci-14-00507-f003:**
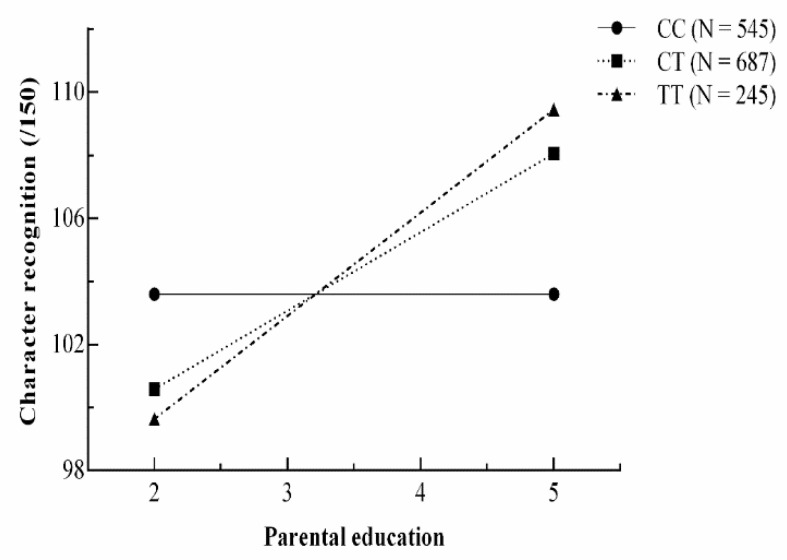
Plots of predicted values as a function of parental education under the strong differential susceptibility gene–environment model for character recognition.

**Table 1 behavsci-14-00507-t001:** Detailed information on the nine selected SNPs.

	SNP	Traits ^a^	Reference	Year	*p* Value ^a^	Base Pair	Risk Allele ^a^	Beta ^a^	Gene	*χ*^2^ (1) ^b^	*p* ^b^
1	rs1541518	Non-word reading	Genome-wide association scan identifies new variants associated with a cognitive predictor of dyslexia [28]	2019	6 × 10^−8^	31,108,665	G	0.177	ADCYAP1R1	0.02	0.99
2	rs281238	Phoneme awareness			1 × 10^−7^	47,432,075	T	0.156	SEMA6D	1.29	0.52
3	rs4571421	Rapid automatized naming of pictures			3 × 10^−7^	188,588,642	C	0.168	LINC02118	0.32	0.85
4	rs7301219	Rapid automatized naming of pictures			5 × 10^−7^	43,731,097	C	0.151	--	2.31	0.31
5	rs9925265	Phoneme awareness			4.51 × 10^−7^	126,496,851	G	0.148	SLC12A3	0.28	0.87
6	rs7187223	Non-word reading	A genome-wide association study for reading and language abilities in two population cohorts [29]	2013	1 × 10^−7^	82,424,128	A	0.251	--	1.58	0.45
7	rs764255	Word reading			1.8 × 10^−7^	72,271,184	T	−0.077	ZFHX3	2.62	0.27
8	rs6963842	Rapid automatized naming of letters	Multivariate genome-wide association study of rapid automatized naming and rapid alternating stimulus in Hispanic American and African-American youth [31]	2019	2 × 10^−7^	107,994,544	G	0.02	LAMB1	0.21	0.90
9	rs9540938	Latent naming speed			5 × 10^−7^	66,867,593	A	--	PCDH9	1.35	0.51

Note: ^a^ Information collected from original studies. ^b^ Results analyzed from current study.

**Table 2 behavsci-14-00507-t002:** Results for alternative regression models for rs281238 on reading ability.

Standard Parameterization			Re-Parameterized Regression Equation				
				Differential Susceptibility		Diathesis–Stress	
Parameter	Gene(G) and Environment(E) Main Effects: Model 1	Main Effects and G×E Interaction: Model 2	Parameter	Strong: Model 3a	Weak: Model 3b	Strong: Model 3c	Weak: Model 3d
*A* _0_	−7.61 (5.67)	−3.71 (5.83)	*C*	3.21 (0.36)	3.21 (0.41)	6.85 (--) ^a^	6.84 (--) ^a^
*A* _1_	1.88 (0.43)	0.63 (0.62)	*A* _0_	−0.90 (4.82)	−1.70 (5.00)	6.21 (4.60)	5.08 (4.60)
*A* _2_	0.01 (0.67)	−5.08 (1.94)	*A* _1_	--	0.43 (0.67)	--	1.77 (0.46)
*A* _3_	--	1.56 (0.56)	*A* _2_	2.49 (0.60)	2.52 (0.60)	0.48 (0.26)	1.89 (0.45)
*A* _4_	0.92 (0.04)	0.92 (0.04)	*A* _3_	3.27 (0.97)	3.30 (0.97)	0.68 (0.36)	2.09 (0.52)
*A* _5_	−2.56 (0.95)	−2.54 (0.95)	*A* _4_	0.91 (0.04)	0.92 (0.04)	0.86 (0.04)	0.92 (0.04)
*R* ^2^	0.2531	0.2566	*A* _5_	−2.53 (0.95)	−2.54 (0.95)	−2.48 (0.95)	−2.56 (0.95)
*F*	126.1	102.9	*R* ^2^	0.2567	0.2569	0.2452	0.2530
*df*	41,472	51,471	*F*	102.90	103.00	120.90	101.00
*p*	<0.0001	<0.0001	*df*	51,471	61,470	41,472	51,471
*F* vs. 1	--	7.81	*p*	<0.0001	<0.0001	<0.0001	<0.0001
*df*	--	51,471	*F* vs. 3b	0.41	--	11.175	7.66
*p*	--	0.0052	*df*	11,470	--	21,470	11,470
			*p*	0.5202	--	<0.0001	0.0057
			*F* vs. 3c	21.95	11.75	--	14.27
			*df*	11,471	21,470	--	11,471
			*p*	<0.0001	<0.0001	--	0.0001
AIC	12,770.55	12,764.73	AIC	12,764.51	12,766.09	12,784.38	12,771.77
BIC	12,802.34	12,801.81	BIC	12,801.59	12,808.47	12,816.17	12,808.85

Note: AIC, Akaike information criterion; BIC, Bayesian information criterion. Tabled values are parameter estimates, with their standard errors in parentheses. *F* vs. 1, 3b, and 3c stands for *F* test of difference in *R*^2^ for Model 2 versus Model 1, Model 3a, 3c, 3d versus Model 3b, and Model 3a, 3b, 3d versus Model 3c, respectively. ^a^ Parameter fixed at reported value; SE is not applicable, so it is listed as (--).

**Table 3 behavsci-14-00507-t003:** Results for alternative regression models for CGS on character recognition.

Standard Parameterization			Re-Parameterized Regression Equation		
				Differential Susceptibility	Diathesis–Stress
Parameter	Gene(G) and Environment(E) Main Effects: Model 3	Main Effects and G×E Interaction: Model 4	Parameter	Model 3e	Model 3f
*A* _0_	−7.05 (6.01)	5.69 (8.41)	*C*	3.34 (0.56)	6.84 (--) ^a^
*A* _1_	1.87 (0.43)	−2.09 (1.88)	*A* _0_	−1.28 (5.02)	5.06 (4.59)
*A* _2_	−0.80 (0.29)	−1.75 (0.82)	*A* _1_	−2.09 (1.88)	1.33 (0.72)
*A* _3_	-	0.52 (0.24)	*A* _2_	0.52 (0.24)	0.07 (0.07)
*A* _4_	0.92 (0.04)	0.92 (0.04)	*A* _3_	0.92 (0.04)	0.92 (0.04)
*A* _5_	−2.56 (0.95)	−2.57 (0.95)	*A* _4_	−2.57 (0.95)	−2.573 (0.95)
*R* ^2^	0.2552	0.2575	*R* ^2^	0.2555	0.2536
*F*	126.10	102.06	*F*	127.70	126.40
*df*	41,472	51,471	*df*	51,471	41,472
*p*	<0.0001	<0.0001	*p*	<0.0001	<0.0001
*F* vs. 4	--	4.68	*F* vs. 3f	3.54	--
*df*	--	11,471	*df*	11,471	--
*p*	--	0.0307	*p*	0.0491	--
AIC	122,770.47	12,767.78	AIC	12,767.78	12,769.67
BIC	12,802.26	12,804.87	BIC	12,804.87	12,801.46

Note: AIC, Akaike information criterion; BIC, Bayesian information criterion. Tabled values are parameter estimates, with their standard errors in parentheses. *F* vs. 4, 3f stands for *F* test of difference in *R*^2^ for Model 4 versus Model 3, and Model 3f versus Model 3e, respectively. ^a^ Parameter fixed at reported value; SE is not applicable, so it is listed as (--).

## Data Availability

The beta values of the SNPs in this paper can be found in previous studies. The statistical methods and statistical data are available from the corresponding author upon reasonable request. The raw data are not publicly available due to legal or ethical restrictions.

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
