# Peer review of "Interaction between Risk Single-Nucleotide Polymorphisms of Developmental Dyslexia and Parental Education on Reading Ability: Evidence for Differential Susceptibility Theory"

_behavsci, 2024, doi:10.3390/bs14060507_

Round 1
Reviewer 1 Report
Comments and Suggestions for Authors
Thank you for the opportunity to review your manuscript. This information is essential, and I appreciate your motivation to examine this issue. I would like to see a paragraph on recommendations for practitioners or for stakeholders who may be able to make a difference in the lives of individuals at risk for dyslexia.
Comments on the Quality of English Language
Your manuscript is well written. However, I found a few grammatical and spelling errors. For example, the word DATA is the plural form of DATUM which requires a verb in the plural. The word AUTOMATIZED is spelled with Z and not S.
Reviewer 2 Report
Comments and Suggestions for Authors
The manuscript entitled: Interaction between risk SNPs of Developmental Dyslexia and Parental Education on Reading Ability: Evidence for Differential-Susceptibility Theory “ by Qing Yang et al., (Manuscript ID: behavsci-2977637) is a GxE study that describes the Cumulative Genetic Score (CGS) of 9 SNPs, related to developmental dyslexia, and the effect of the parental education on reading ability in 1477 school-aged children. This is a rather well written MS which may be of interest to readers, especially those focused on the molecular basis of the development of dyslexia. I have no additional comments for the authors.
